# Loss of *rpoE* Encoding the δ-Factor of RNA Polymerase Impacts Pathophysiology of the *Streptococcus pyogenes* M1T1 Strain 5448

**DOI:** 10.3390/microorganisms10081686

**Published:** 2022-08-22

**Authors:** Joseph S. Rom, Yoann Le Breton, Emrul Islam, Ashton T. Belew, Najib M. El-Sayed, Kevin S. McIver

**Affiliations:** Department of Cell Biology and Molecular Genetics, Maryland Pathogen Research Institute, University of Maryland, College Park, MD 20742, USA

**Keywords:** *Streptococcus pyogenes*, GAS, *rpoE*, virulence, RNA-seq, *arcA*

## Abstract

*Streptococcus pyogenes*, also known as the Group A Streptococcus (GAS), is a Gram-positive bacterial pathogen of major clinical significance. Despite remaining relatively susceptible to conventional antimicrobial therapeutics, GAS still causes millions of infections and hundreds of thousands of deaths each year worldwide. Thus, a need for prophylactic and therapeutic interventions for GAS is in great demand. In this study, we investigated the importance of the gene encoding the delta (δ) subunit of the GAS RNA polymerase, *rpoE*, for its impact on virulence during skin and soft-tissue infection. A defined 5448 mutant with an insertionally-inactivated *rpoE* gene was defective for survival in whole human blood and was attenuated for both disseminated lethality and lesion size upon mono-culture infection in mouse soft tissue. Furthermore, the mutant had reduced competitive fitness when co-infected with wild type (WT) 5448 in the mouse model. We were unable to attribute this attenuation to any observable growth defect, although colony size and the ability to grow at higher temperatures were both affected when grown with nutrient-rich THY media. RNA-seq of GAS grown in THY to late log phase found that mutation of *rpoE* significantly impacted (>2-fold) the expression of 429 total genes (205 upregulated, 224 downregulated), including multiple virulence and “housekeeping” genes. The *arc* operon encoding the arginine deiminase (ADI) pathway was the most upregulated in the *rpoE* mutant and this could be confirmed phenotypically. Taken together, these findings demonstrate that the delta (δ) subunit of RNA polymerase is vital in GAS gene expression and virulence.

## 1. Introduction

*Streptococcus pyogenes*, also known as Group A Streptococcus (GAS) causes hundreds of millions of non-invasive infections every year, often in the upper respiratory tract, but also has the ability to cause more invasive diseases, such as necrotizing fasciitis and sepsis that account for some of the 500,000 deaths caused by GAS every year [1]. Despite remaining sensitive to first-line antibiotics, such as penicillin [2], the sheer volume of these infections still presents a major burden to healthcare systems around the world. Additionally, there have been recent reports of the development of resistance to antibiotics, such as clindamycin and tetracycline, which are necessary therapeutic options for those with penicillin allergies [3,4,5]. Therefore, it is important to gain insights into the biology of GAS in order to develop novel treatments to mitigate persistent GAS infections.

When considering therapeutic drug candidates, the goal should be to selectivity target a bacterial cell process/molecular component that is not present in the host. Unlike eukaryotes, prokaryotic RNA polymerases (RNAPs) contain certain unique components, notably essential and non-essential σ-factors [6]. Indeed, there are already many clinically-approved antimicrobials that disrupt bacterial translation by interfering with the bacterial ribosome, but to date, there are only two drugs that target bacterial transcription (rifamycin and fidaxomicin/lipiarmycin) [7]. Thus, targeting components of the RNAP-subunits is a highly underutilized strategy for developing new antimicrobial drugs.

In addition to its core components, prokaryotic RNAP holoenzymes contain a variety of non-essential cofactors, notably the delta-subunit or δ-factor that is encoded by the gene *rpoE* and only found in Gram-positive bacteria [8]. σ-factors are thought to work in tandem with the δ-factor, whereby the former enhances specific RNAP affinity through direct interaction with DNA [9,10], and the latter decreases binding affinity by either directly or indirectly lowering the affinity of the RNAP to cis-regulatory elements [8,11,12,13,14]. This results in the δ-factor reducing RNAP binding to low-affinity promotor regions while having little impact on the binding to high-affinity promotor regions. Weiss and Shaw provide a detailed review of the δ-factor (*rpoE*) and other accessory RNAP subunits in Gram-positive bacteria [15].

A number of research groups have studied the effect of deleting *rpoE* on a diverse set of Gram-positive bacteria. For instance, mutation of *rpoE* in *Staphylococcus aureus* resulted in impaired viable cell recovery when grown in nutrient-limiting environments [16] whereas in *Streptococcus mutans*, as well as *Streptococcus agalactiae*, it led to an increased lag-phase [17,18]. In *Bacillus subtilis*, mutating *rpoE* also increases the lag-phase of growth [19], and this delay was thought to account for decreased competitive fitness observed when mutant bacteria are cocultured in vitro with WT bacteria [20]. Mutation of *rpoE* also appears to affect virulence in pathogenic Gram-positive bacteria by limiting the ability of *S. aureus* and *S. agalactiae* to resist phagocytic killing and reducing disease severity in rodent models of systemic infection [17,21,22,23]. However, mutating *rpoE* might have different consequences for pathogenesis depending on the pathogen in question, as Xue and colleagues found that loss of δ-factor in *S. mutans* increased bacterial attachment to human extracellular matrix components [24]. Nevertheless, it is clear that *rpoE* plays an important role in diverse regulatory functions and cell homeostasis.

To date, the δ-factor (*rpoE*) has not been investigated in GAS. In this report, we sought to understand the importance of *rpoE* in GAS pathogenesis and whether it shares some of the same biological roles observed in other Gram-positive bacteria. We found that mutation of *rpoE* significantly limits GAS virulence and fitness during murine skin and soft-tissue infection but did not affect bacterial replication unless grown at elevated temperatures. Additionally, RNA-seq on the *rpoE* mutant revealed global alterations in gene expression, which included the arc operon encoding the arginine deiminase (ADI) pathway. Our data suggest mutation of *rpoE* leads to disruption of fundamental cell processes that in turn may reduce the bacteria’s ability to cope with stressful environments, such as those found in the host.

## 2. Materials and Methods

### 2.1. Bacterial Strains and Growth Conditions

The GAS strains used in these experiments were all generated in the M1 serotype strain 5448 and are summarized in Table 1 [25,26,27,28,29]. All bacterial strains were stored at −80 °C in THY containing 20% (*v/v*) glycerol. For each study, strains were recovered from the freezer and plated to TSA with 5% sheep blood or THY agar plates containing the appropriate antibiotic selection. Antibiotics were used in the following concentrations: ampicillin 100 μg mL^−1^; spectinomycin, 100 μg mL^−1^; erythromycin, 0.5 μg mL^−1^. Overnight cultures were prepared from ≤24 h old plates and incubated at 37 °C for 14–16 h prior to use.

### 2.2. Insertional Mutagenesis and Complementation

The *covS*, M5005_Spy_1849, and the *arcA* isogenic mutants of GAS M1T1 5448 were previously generated using the pSinS suicide vector and the temperature-sensitive pHelpK plasmid as described [27,28]. The *rpoE* mutant was generated in 5448 using the same approach. Briefly, an internal fragment of *rpoE* was PCR amplified using the primers CCC GGA TCC GAC AAG AAA AAA GCG AGC TTT CC (oll1611F), and CCC GGA TCC CGT TTT TTC TTA CGT TTT TGT GC (oll1611R), and then cloned into the pSinS. The resulting suicide construct was then transformed into 5448 containing the pHlpK helper plasmid and an insertion was facilitated using temperature shift as previously described [31].

To generate a complemented version of the *rpoE* mutant, a 1152 bp region including the entire *rpoE* ORF as well as ca. 400 bases upstream was amplified from WT 5448 gDNA using the primers aaa gag ctc GCA ATT GCT AAA GCA GAA GG (RpoE comp F) and aaa gga tcc GGC AAA CTG GCA TTA TTC TAG (RpoE comp R). This fragment was cloned into the pCR Blunt II TOPO vector (Invitrogen, Waltham, MA, USA) and transformed into TOP10 *E. coli* (Invitrogen). Once confirmed, the plasmid was re-isolated and digested with BamHI and XhoI and cloned into the pAMβ1 origin-containing vector pOri253 for replication in GAS. This construct was transformed into 5448 *rpoE* mutant cells and selected on THY agar plates containing 0.5 μg/mL^−1^ erythromycin.

### 2.3. Murine Subcutaneous Infection Model

The model was used as previously described [28]. Briefly, 5448 strains were grown overnight and diluted 1:20 into THY broth to a final volume of 80 mL and grown to the late-logarithmic phase. To ensure homogenous inoculum, GAS cell suspensions were vortexed for 10 min, and the cells were then centrifuged at 6000× *g* for 10 min and resuspended in saline to produce the infection inoculum. Initial CFU counts were obtained by use of a Petroff-Hauser slide and the counts were confirmed by serially drop dilutions on THY agar plates. Five-week-old, outbred, immunocompetent, hairless female Crl:SKH1-hrBR mice (Charles River Laboratories) received subcutaneous injections in their backs. Moribund mice were euthanized by CO_2_ asphyxiation or at the end of the study at 7 d. Skin lesions were measured for the area (cm^2^) at 48 h post-infection, depending on the experiment. For the survival study, a Mantel-Cox test was used to evaluate the significance of differences between the WT and *rpoE* mutant groups.

### 2.4. In Vivo Competition Assay

Co-infections with WT 5448 and isogenic mutant strains (*rpoE*, *covS*, M5005_Spy_1849) were carried out using 5- to 6-week-old female CD-1 outbred mice (Charles River Laboratories) as described elsewhere [28]. The *covS* and M5005_Spy_1849 results have been previously published [28]. Briefly, cell suspensions of exponentially growing GAS cells were obtained by mixing equal amounts of GAS 5448 and an isogenic mutant strain (~1:1) in saline (10^9^ CFU/mL). Mice were anesthetized with ketamine, fur was removed from a ~3 cm^2^ area of the haunch with Nair (Carter Products), and 100 μL of a cell suspension in saline injected under the back skin. Mice were monitored twice daily for 2 days and euthanized by CO_2_ asphyxiation. Skin lesions lysates were harvested in a sterile 2-mL screw-cap microtube containing 1.4 mm ceramic spheres (Lysing Matrix D, MP Biomedicals, Santa Ana, CA, USA) and 1 mL of sterile saline, and skin tissues were homogenized using three successive 45-s bursts with a FastPrep FP120 BeadBeater (BioSpec Products, Bartlesville, OK, USA). Tissue homogenates (3 mL final) were serially diluted (10-fold increments) in saline, plated on either THY (whole population) or THY containing the appropriate antibiotic (mutant population), and cell counts were determined. The competition index (CI) was calculated using the following formula: CI = (R_M_/R_W_)/(R_M0_/R_W0_), with R_M0_ and R_W0_ corresponding to the ratio of the mutant and the ratio of the wild type, respectively, in the initial inoculum (T_0_); and R_M_ and R_W_ corresponding to the ratio of the mutant and the ratio of the wild type, respectively, at the end of the competition growth assay. Unpaired student’s *t*-test was used to evaluate the significance of differences between groups; a *p* value of <0.05 was considered statistically significant.

### 2.5. Lancefield Bactericidal Assay

Bacterial survival was measured as previously described [32]. Overnight GAS cultures were used to inoculate fresh THY cultures and grown to the mid-logarithmic phase. Cultures were then serially diluted in saline and 50 μL of the 10^−4^ dilution was used to inoculate 500 μL of fresh whole human blood, which was incubated while rotating at 37 °C. After 3 h, cultures were serially diluted and plated on THY agar. The multiplication factor (MF) was determined by dividing the CFU obtained after growth in blood by the initial CFU inoculum. Data are then presented as percent growth in blood using the following formula: percent growth = (MF of mutant/MF of WT) × 100. Unpaired student’s *t*-test was used to evaluate the significance of differences between groups; a *p* value of <0.05 (*) was considered statistically significant.

### 2.6. In Vitro Growth Studies

To test bacterial growth, overnight cultures of GAS were prepared in fresh THYB with erythromycin (0.5 μg mL^−1^) and incubated overnight in closed tubes at 37 °C for 14–16 h. Bacteria were then centrifuged and washed once with sterile saline to remove the spent media and then reconstituted in fresh media at a standardized optical density of 600 nm (OD_600_) = 0.05 in sterile Klett tubes. For growth in nutrient-rich conditions, the bacteria were reconstituted in fresh THY with erythromycin (0.5 μg mL^−1^). For growth in nutrient-limiting conditions, the bacteria were reconstituted in mRPMI. This media was prepared by mixing phenol-red-free RPMI medium 1640 (Gibco, Waltham, MA, USA; 32404-014) with L-glutamine (200 mM), adenine (12.5 mg mL^−1^), Uracil (50 mg mL^−1^), guanine (2.5 mg mL^−1^), 1× diluted BME vitamins (Sigma, St. Louis, MO, USA; B6891), glucose (20% *w/v*), and HEPES (50 mM; pH = 7.4). Bacterial growth was measured based on absorbance readings using a spectrophotometer. Each experiment was performed in three biological replicates, each containing three technical replicates.

In order to measure growth at higher temperatures, bacterial overnights were prepared as described above, but 120 μL of each fresh bacterial suspension was aliquoted into six PCR tubes; one tube for each temperature reading. The tubes were then placed on a thermocycler and incubated at a temperature gradient from 37.1 °C to 39.6 °C. This experiment was carried out in six separate biological replicates.

### 2.7. RNA-Seq and Data Analysis

RNA sequencing (RNA-Seq) was performed as previously described [33]. Briefly, Direct-zol RNA MiniPrep kit (Zymo Research, Irvine, CA, USA) was used to isolate total RNA using a modified procedure to improve GAS cell disruption. Frozen cell pellets were resuspended in 700 µL of TRIzol plus 300 mg of acid-washed glass beads (Sigma Life Science, St. Louis, MO, USA) and disrupted by vortexing for 5 min. RNA samples were DNase-treated using the Turbo DNase-free kit (Life Technologies, Carlsbad, CA, USA). A total of 5 µg of this RNA was treated for ribosomal RNA (rRNA) removal using the Ribo-Zero Magnetic kit for Gram-positive bacteria (Epicentre, San Diego, USA). Quality and quantity were assessed using a 2100 Bioanalyzer (Agilent, Santa Clara, CA, USA) NanoDrop 8000 spectrophotometer (Thermo Scientific, Walthamm, MA, USA), respectively. Directional RNA-Seq libraries were created using the ScriptSeq v2 RNA-Seq Library Preparation kit (Illumina, San Diego, CA, USA) according to the manufacturer’s recommendations. The libraries were sequenced at the Institute for Bioscience and Biotechnology Research (IBBR) Sequencing Facility at the University of Maryland, College Park; reads were deposited at the Sequence Read Archive under accession PRJNA412519.

Read quality was measured using FastQC, filtered and trimmed using trimmomatic [34], and mapped against the MGAS5005 genome (accession CP000017) using hisat2 [35]. Differential expression analyses were performed following visualization of size-factor and quantile normalization of read counts; potential batch effects were evaluated by comparing the DESeq2 results under three conditions: without any additional parameters to the experimental model, the addition of the most likely batch factor to the model, and after adding sva-derived estimates to the model [36]. Differential expression analyses were performed with DESeq2 [37], and a statistically uninformed basic method as a negative control. The resulting metrics of expression were tested for ontology enrichment using goseq [38].

### 2.8. Extracellular Acidification Assay

Bacterial acidification was measured using a protocol adapted from a published study [39]. Briefly, acidification media (AM) was made by supplementing THY with 18 μg mL^−1^ of phenol red and 30 mM arginine. GAS overnight cultures grown in THY were used to inoculate AM at a starter to a fresh media ration of 1:20. Acidification was measured at 4, 6, 8, 10, and 24 h. At each time point, quantitative measurements were done by removing bacterial cells via centrifugation (15,000× *g* for 3 min) and measuring AM using a spectrophotometer (Genesys 30, Thermo Scientific) at an optical density of 560 nm. This experiment was performed in three separate biological replicates with a representative experiment presented in this publication. Unpaired student’s *t*-test was used to evaluate the significance of differences between groups; a *p* value of <0.05 (*) was considered statistically significant.

### 2.9. Ethics Statement

All mice were infected with GAS in AAALAC-accredited ABSL-2 facilities and following protocols approved by the University of Maryland IACUC (R-16-05) for humane treatment of animal subjects in accordance with guidelines set up by the Office of Laboratory Animal Welfare at NIH, Public Health Service, and the Guide for the Care and Use of Laboratory Animals; with every effort to limit distress and pain to animals taken. Human blood donations were approved by the University of Maryland Institutional Review Board (10-0735), and the written consent of donors was archived.

## 3. Results

### 3.1. An rpoE Mutant in the M1T1 GAS Strain 5448 Is Attenuated for Virulence

Given the impact that mutation of *rpoE* has shown on altering virulence in Gram-positive bacterial pathogens [21,22,24], we sought to determine how mutation of *rpoE* might affect virulence in GAS. We generated a mutant lacking a functional *rpoE* gene in the M1T1 strain 5448, a clinically relevant strain of GAS with a propensity to cause invasive infections [26]. The isogenic *rpoE* mutant was created using the stable pSinS/pHlpK insertional inactivation system developed by our group [28]. Both wild type (WT) 5448 and the *rpoE* mutant strains were inoculated subcutaneously into the hind flanks of 5–7-week-old mice in order to model skin and soft-tissue infection, and both lethality and lesion size at the site of inoculation were measured over seven days (Figure 1A,B). All mice infected with WT 5448 succumbed by day 5, whereas all but one mouse from the group infected with the *rpoE* mutant survived the seven-day time course of the experiment. Additionally, the lesion sizes (cm^2^) of the mice infected with the *rpoE* mutant were significantly smaller than those of WT, indicating that an intact *rpoE* locus is required in order for GAS to maintain virulence during soft-tissue infection in vivo.

### 3.2. The 5448 WT Strain Outcompetes the rpoE Mutant When Co-Infected in the Skin and Soft-Tissue Infection Model

In *B. subtilis*, the loss of the δ-factor resulted in a reduction of competitive fitness when co-cultured with the parent strain in vitro [20]. Therefore, we sought to determine if a similar phenotype was apparent with our *rpoE* mutant in GAS. Our group had previously conducted a Tn-seq study to identify genes that were required for virulence in murine soft-tissue infection [28]. Although the *rpoE* gene was not identified in the Tn-seq [28], we still included our *rpoE* mutant in the competition experiments co-infected with WT 5448 in the soft-tissue model (unpublished). The *rpoE* mutant was outcompeted by WT 5448, indicating a significantly decreased level of fitness in vivo (Figure 1C). For comparison, a mutant in a putative Zn^2+^ metalloprotease (M5005_Spy_1849) showed no impact on fitness and a mutant in *covS* exhibited significantly enhanced fitness, both published but from the same experiment performed with *rpoE* [28]. Thus, mutation of *rpoE* reduces the ability of GAS to proliferate inside the host.

### 3.3. Mutation of rpoE Limits the Ability of GAS 5448 to Survive in Human Blood

Mutation of *rpoE* in both *S. aureus* and *S. agalactiae* was found to reduce their survival in human blood [17,22]. Given the evolutionary divergence between mice and humans, we sought to gain a better understanding of how mutation of *rpoE* might impact the ability of GAS to evade the human innate immune system during systemic infection. To test this, we performed a Lancefield bactericidal assay on the WT 5448 and the *rpoE* mutant. Mutation of *rpoE* significantly reduced GAS recovered after 3 h incubation in non-immune human blood by comparison to the WT 5448 (Figure 2). These findings are consistent with the significant decrease in lethality and wound severity in mice caused by the *rpoE* mutant, indicating significant attenuation in both mouse and human models of infection.

### 3.4. rpoE Mutants Produce a Small Colony Variant Phenotype, Enhanced Aggregation, and No Changes in Growth Kinetics Compared to WT

Previous studies in *S. agalactiae* (Group B streptococcus, GBS) showed that a mutant of *rpoE* grew, such as the parent strain in chemically defined media in vitro, but its growth was markedly reduced when grown in nutrient-rich Todd-Hewitt Yeast broth or THYB [17]. In *S. pneumoniae*, mutation of *rpoE* resulted in a small-colony variant (SCV) phenotype when the mutant was grown on Columbia Blood agar [40], while in *S. mutans* a *rpoE* mutation led to enhanced bacterial [18]. Both the SCV and clumping phenotypes could be associated with delayed cell replication, which might in turn account for the attenuation observed for the GAS mutant in vivo (Figure 1 and Figure 2).

Therefore, we assessed the relative ability of an *rpoE* mutant to grow using different solid and liquid growth media (Figure 3). When grown on tryptic soy agar with 5% sheep’s blood, GAS formed semi-translucent grey colonies with characteristic β-hemolysis. However, the *rpoE* mutant produced smaller, more opaque white colonies that retained β-hemolytic activity (Figure 3A,B). Genetic complementation of the *rpoE* mutant restored the colony morphology to that of the WT. Interestingly, when grown on THY agar, the *rpoE* mutant exhibited a larger colony phenotype than WT, which was also reversed in the complemented strain. In addition, mutation of *rpoE* significantly enhanced the rate of sedimentation of exponential-growth phase bacteria, and this was apparent only 10 min after bacterial resuspension (Figure 3C). There was no significant difference in sedimentation between the WT and the *rpoE* mutant after 20 min of settlement or the complemented mutant at either 10- or 20-min post-resuspension. These findings raised the possibility that mutation of *rpoE* might result in a growth defect in GAS when cultured in vitro.

To test this, the WT, *rpoE* mutant, and the complemented strain were cultured in liquid media and growth assessed by absorbance at 600 nm over time (Figure 3D,E). To ensure complementation in standardized growth conditions, we transformed the WT and *rpoE* mutant strains with the erythromycin-resistant vector pMSP3535. Interestingly, when grown in either nutrient-rich THY or nutrient-limiting modified RPMI media, the *rpoE* mutant grew with comparable kinetics to the WT, showing only a slight decrease in growth rate in the modified RPMI medium (Figure 3D,E).

### 3.5. Mutation of rpoE Decreases Fitness at High Temperatures

Two previous studies found that mutation of *rpoE* reduced the fitness of *S. aureus* and *S. mutans* when each was grown under specific stressors, such as low pH or H_2_O_2_; however, the impact of these stressors was modest and not shown to be statistically significant [16,18]. We tested the ability of our GAS *rpoE* mutant to resist the effects of H_2_O_2_ in culture media but were unable to observe any reproducible impact in viability compared to WT 5448 (data not shown). Given that loss of *rpoE* may decrease the bacteria’s ability to cope with other environmental stresses, we assessed the impact of increasing temperatures on bacterial growth. The erythromycin-resistant WT 5448 (pMSP3535), the *rpoE* mutant (pMSP3535), and the complemented mutant (pMSP3535 with *rpoE* allele) were grown from a low optical density (OD_600_ = 0.05) in fresh THY media plus erythromycin in a thermocycler at a gradient set to six different temperatures ranging from 37.1 to 39.6 °C (Figure 4). Although higher temperatures reduced the viability of each strain of GAS, at the highest temperature of 39.6 °C, the WT and complemented mutant was still able to grow to turbidity roughly equal to half of that measured at the lowest temperature of 37.1 °C. In contrast, the *rpoE* mutant growth was significantly limited at the highest temperatures of 39.2 °C and 39.6 °C. There was no significant difference in growth between the WT and the complemented mutant at any temperature.

### 3.6. RNA-Seq Shows That Loss of rpoE Alters Global Gene Expression

To better understand the impact of *rpoE* on gene expression in GAS, RNA-seq was performed on the 5448 WT and the *rpoE* mutant. Total RNA was extracted from four biological replicate cultures of WT 5448 and the *rpoE* mutant grown in THY broth to mid-exponential phase and rRNA-depleted samples subjected to RNA sequencing. Under the conditions used, 429 total genes (ca. 25% of the genome) were differentially expressed ≥2 fold in the *rpoE* mutant versus the WT, with 205 genes upregulated and 224 genes downregulated in the absence of *rpoE* (Appendix A). Using gene ontology (GO) analyses of the two-fold differentially-regulated genes, we found that most were involved in processes related to basic cell metabolism. Notably, genes implicated in translation were upregulated whereas a number of phospho(enol)pyruvate phosphotransferase system (PTS) genes involved in carbohydrate uptake were downregulated (Figure 5A,B).

### 3.7. Mutation of rpoE Enhances Arc Operon Expression during Growth of GAS in Liquid Culture Media

Unsurprisingly, mutation of *rpoE* causes dysregulated expression of a large percentage of the GAS transcriptome. A number of virulence genes (e.g., *trxR*/*S* and *norA*), as well as those involved directly in metabolic processes (e.g., *celA*/*B* and *artP*/*Q*), were among the genes most upregulated in the *rpoE* mutant. Of the 205 upregulated genes, the *arc* operon encoding the components of the arginine deaminase system (ADI) pathway were some of the most highly expressed (Figure 5C). The *arc* operon is important for maintaining pH homeostasis and consists of four elements: an arginine deaminase (*arcA*) that creates ammonia by degrading arginine into citrulline, an ornithine carbamoyl transferase (*arcB*) that degrades citrulline into carbamoyl phosphate and ornithine, a carbamate kinase (*arcC*) that makes ATP and ammonia from ADP and carbamoyl phosphate, and finally an arginine/ornithine antiporter (*arcD*) that initiates the uptake of arginine used in this pathway [41,42]. In the absence of preferred energy in the form of glucose, GAS relies on arginine for energy and for maintaining virulence during colonization and early infection [39]. Notably, inactivating the ADI pathway via mutation of *arcA* resulted in the inability of GAS to de-acidify its growth media supplemented with arginine [39].

We hypothesized that the mutation of *rpoE* might result in a more rapid deacidification of its growth media supplemented with arginine due to the upregulation of the *arc* operon. To test this, we grew the WT, the *rpoE* mutant, and the complemented strain in THY supplemented with phenol red and additional arginine (Figure 6). Phenol red can be used quantitatively to measure pH spectrophotometrically at a wavelength of 560 nm with absorbance increasing non-linearly as pH values increase. An isogenic *arcA* mutant was included as a deacidification negative control. Variability led to difficulty in producing similar reacidification kinetics. Still, the *rpoE* mutant consistently re-acidified its media faster than the WT and complemented *rpoE* mutant in multiple independent experiments. Importantly, after 24 h of incubation, all strains, except for the *arcA* mutant, had de-acidified their growth media (purple) whereas the *arcA* mutant media remained acidic (yellow) (Figure 6). It is important to note that the growth kinetics of the *arcA* mutant were slower than that observed with the other strains; however, comparable growth was observed in all culture tubes after 24 h. Taken together, these results phenotypically validate the RNA-seq findings for repression of the *arc* operon by the δ-factor.

## 4. Discussion

GAS has remained generally susceptible to most therapeutic interventions, most notably penicillin; however, there have been reports observing resistance to other drugs, such as clindamycin and tetracycline [3,4,5]. Indeed, antibiotic resistance in many other pathogenic bacteria is on the rise, therefore developing therapeutics to replace ineffective antibiotics is a high priority. When developing a new antimicrobial, the fundamental consideration is to selectively target bacterial cell components that are absent in the host. The δ-factor (*rpoE*) represents one such target. Here, we investigated the importance of the accessory δ-factor of the RNAP in GAS pathogenesis. Using an insertional mutation in *rpoE* (M5005_Spy_1611) [28], we demonstrated that the mutation of *rpoE* attenuated virulence of GAS in a murine skin and soft-tissue model of infection and limited viability in human blood.

The GAS is capable of causing disease in many areas of the body [43]. Typically, these infections are local and self-limiting, however, occasionally, these infections become systemic and/or life-threatening in nature. The GAS M1T1 serotype has a propensity to cause such serious diseases and differs at the genetic level from other less invasive GAS serotypes [44]. For this study, we chose to use the GAS strain 5448, as it is a clinically relevant and globally-disseminated M1T1 serotype known for causing skin and soft-tissue infections [28]. Mice that were mono-culture infected with WT 5448 all succumbed within 7 days, demonstrating that our skin and soft-tissue infection model facilitated lethal disseminated disease. In contrast, all but one mouse infected with the *rpoE* mutant survived the time-course of the experiment, and these animals all displayed smaller lesions at the site of inoculation (data not shown). Mice co-infected with a lower dose inoculum of GAS containing a ~50:50 cell suspension of WT and the *rpoE* mutant favored the growth of WT bacteria in infected skin tissue.

Our mouse infection experiments suggested the possibility that the mutation of *rpoE* reduces the ability of the bacteria to evade innate host defenses. This was supported by the Lancefield bactericidal results using whole human blood. Mutation of *rpoE* significantly limited the ability of GAS to survive in human blood by comparison to the WT. This is consistent with two previous studies that used δ-factor defective strains in *S. aureus* or *S. agalactiae* in whole human blood [17,22]. In *S. aureus*, Weiss and colleagues took their studies a step further and also demonstrated that their *rpoE* mutant was more susceptible to isolated leukocytes than the WT. Taken with our findings, increased phagocyte susceptibility of the *rpoE* mutant may at least in part account for its attenuation in vivo.

Previous reports have shown that mutation of *rpoE* can increase the lag phase of growth [17,19,22]. These observations raised the possibility that the attenuation and competition defect observed in our *rpoE* mutant could be directly attributed to a growth defect. When grown in nutrient-rich THYB, we found no evidence of a growth defect in the *rpoE* mutant by comparison to the WT. However, there was a very slight delay in logarithmic growth when the *rpoE* mutant was grown in nutrient-limited modified-RPMI media, thus presenting the possibility that even a small growth defect in vitro may have biological significance when the bacteria are replicating in a host. It is worth noting that fresh plates were struck out for each growth experiment and the incubation time for our starter cultures prior to subculturing fresh media was consistent (14–16 h for each replicate). Thus, we cannot rule out the possibility that mutation of *rpoE* delays growth when the bacteria are recovered from a nutrient-starved culture.

The most readily apparent phenotypic change in the GAS *rpoE* mutant was a small-colony variant (SCV) phenotype when grown on TSA w/5% sheep’s blood plates. Interestingly, we saw what appeared to be larger, more diffuse mutant colonies compared to WT when grown on THY agar. Additionally, when grown in liquid THY, the *rpoE* mutant tended to settle out of the solution faster than the WT or the complemented mutant. These changes are consistent with previous reports that found SCV and clumping phenotypes associated with *rpoE* mutations in *S. pneumoniae* and *S. mutans*, respectively [18,40]. Observing both of these phenotypic changes in our GAS *rpoE* mutant indicate changes in global gene expression, especially those genes involved with cell replication and adherence.

Multiple reports have shown varying effects on the viability of an *rpoE* mutant when exposed to low pH and H_2_O_2_ stress [16,18]. However, growing GAS at hyper-optimal temperatures negatively impacted bacterial growth in the *rpoE* mutant to a degree noticeably higher than that of the WT or the complemented mutant. These results indicate that the *rpoE* mutant might be less able to carry out a protein-unfolding stress response due to global alterations in its gene expression.

Since the δ-factor is a component of the RNA polymerase, it is not surprising that mutants in *rpoE* often demonstrate pleiotropic effects on gene expression [22]. In our RNA-seq analysis of the 5448 *rpoE* mutant compared to WT grown in THY, we observed a broad impact (428 total genes, ca. 25% of the genome) on the GAS transcriptome. This included the upregulation of numerous metabolic genes as well as virulence factors in the *rpoE* mutant. Notably, the arc operon genes, which are critical in order for GAS to invade the host when colonizing the skin [39], were most highly upregulated in our *rpoE* mutant. We provided phenotypic confirmation that the arc operon was indeed more highly expressed in the *rpoE* mutant, as the mutant was able to de-acidify its growth media in the early stationary phase faster than either the WT or the complemented mutant. We also found that an *arcA* mutant was not able to de-acidify its culture media after 24 h, which was consistent with a previous study [39]. However, Hirose and colleagues also showed that an *arcA* mutant was attenuated in mouse skin infection and this was attributed to an inability of the mutant to acquire alternative energy in the form of filaggrin-derived arginine [39]. Therefore, it is hard to attribute the *rpoE* mutant attenuation in the skin and soft-tissue infection being due to alterations in the ADI pathway. Importantly, the *trxR*/*S* two-component system operon was also highly upregulated in the 5448 *rpoE* mutant. A previous study found that upregulation of *trxS* enhances adhesion and biofilm production which might account for the increased sedimentation that we had observed in our mutant [45]. It is uncertain whether this phenotype on its own could account for enhanced GAS-host cell adhesion, although any increase in GAS cell adherence, such as that we observed in the bacterial clumping assay, would fail to explain the attenuation of the *rpoE* mutant we observed during skin and soft-tissue infection.

In *S. mutans*, Xue and colleagues showed that the mutation of *rpoE* led to decrease the sensitivity to environmental stress [18]. Using MALDI-TOF mass spectroscopy, they later discovered that the mutation altered the abundance of metabolic and stress adaptation proteins leading them to conclude that the role of *rpoE* on stress responses was more likely due to global changes in “housekeeping” genes rather than to any specific mechanism [46]. In GAS 5448, we found that many of the genes downregulated in the *rpoE* mutant were involved in translation machinery (e.g., 30S and 50S ribosomal subunits) and specifically carbohydrate metabolism. Taken together with our experiments, we hypothesize that global alterations in gene expression in the GAS *rpoE* mutant likely reduce its ability to cope with environmental stress (e.g., high temperatures and phagocytosis), which in turn may account for its decreased competitive fitness when co-infected with the 5448 WT and virulence in vivo. In conclusion, these findings emphasize the importance of the δ-factor in GAS pathogenesis and maintaining intracellular homeostasis. Importantly, they highlight the *rpoE* gene itself or the δ-factor as potentially viable therapeutic targets. Future studies will be needed to determine if the impacts of *rpoE* on virulence and physiology are conserved in other GAS serotypes.

## Figures and Tables

**Figure 1 microorganisms-10-01686-f001:**
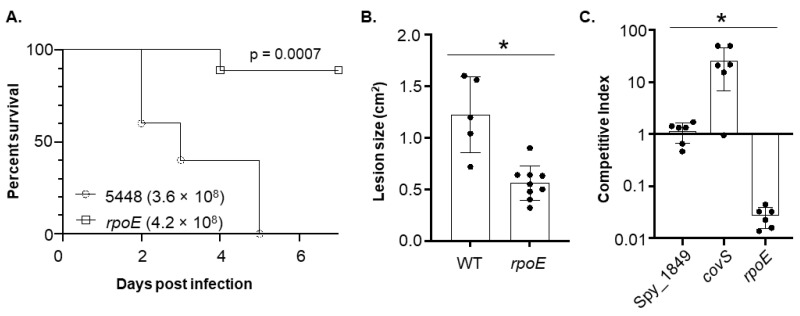
**Mutation of *rpoE* in 5448 results in attenuation and loss of fitness in a murine soft-tissue infection model.** (**A**) Disseminated systemic lethality was monitored following subcutaneous infection with wild type (WT, n = 5) 5448 or the *rpoE* mutant (*rpoE*, n = 9) at the CFU doses indicated. The *p* value indicates statistical significance in lethality between the *rpoE* mutant and WT. (**B**) A separate experiment was done to measure ulcerative lesion size following subcutaneous infection after 48 h post-inoculation (pi). (**C**) Competition assays of a Spy_1849 mutant, a *covS* mutant (*covS*) and a *rpoE* mutant each co-infected with an equivalent inoculum of WT 5448 in the mouse model. The control data for Spy_1849 and *covS* were published [28] whereas the *rpoE* mutant was assayed in the same experiment but not previously published. CFU was harvested 48 h p.i. and quantified by plate dilution. Asterisk indicates statistical significance (*p* ≤ 0.05) compared to WT (**B**) and each strain (**C**) using an unpaired *t*-test.

**Figure 2 microorganisms-10-01686-f002:**
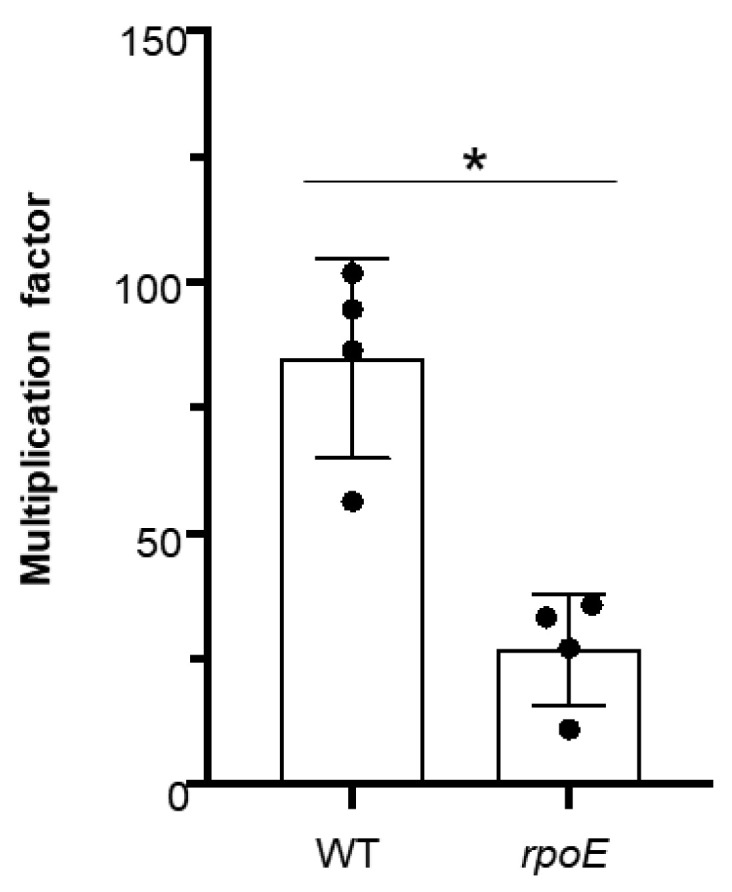
**Mutation of *rpoE* in M1T1 5448 reduces survival in human blood.** A Lancefield bactericidal assay in non-immune whole human blood was performed using WT 5448 (WT) and a *rpoE* mutant (*rpoE*). Asterisk indicates statistical significance (*p* ≤ 0.05) by comparison to the WT as determined by an unpaired *t*-test.

**Figure 3 microorganisms-10-01686-f003:**
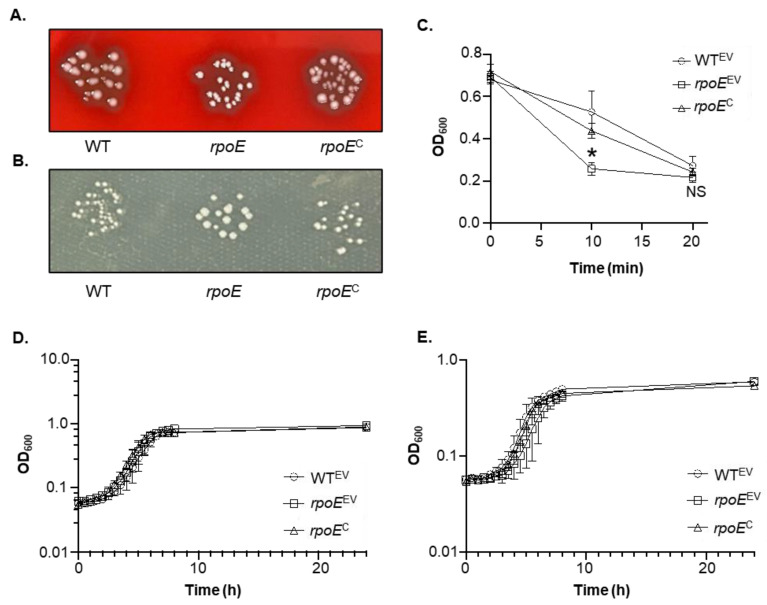
**The *rpoE* mutant alters *S. pyogenes* colony morphology without impacting growth.** (**A**,**B**) Serial dilutions were performed from overnight cultures of the wild type (WT), *rpoE* mutant (*rpoE*), or the complemented *rpoE* mutant strain (*rpoE*^C^), and dilutions spotted on tryptic soy agar with 5% sheep’s blood (**A**) or Todd Hewitt Yeast agar (**B**) and incubated at 37 °C with 5% CO_2_ overnight. (**C**) Cultures were grown to exponential growth and then turbidity was measured over time at an optical density of 600 nm (OD_600_) after tube inversion to assess aggregation. (**D**,**E**) Bacteria were grown from a starting OD_600_ = 0.05 either in nutrient-rich THY broth (**D**) or nutrient-limiting modified RPMI (**E**) for 24 h and OD_600_ was recorded over time. Asterisk indicates statistical significance (*p* ≤ 0.05) of the *rpoE* mutant by comparison to the WT as determined by an unpaired *t*-test. NS = not significant.

**Figure 4 microorganisms-10-01686-f004:**
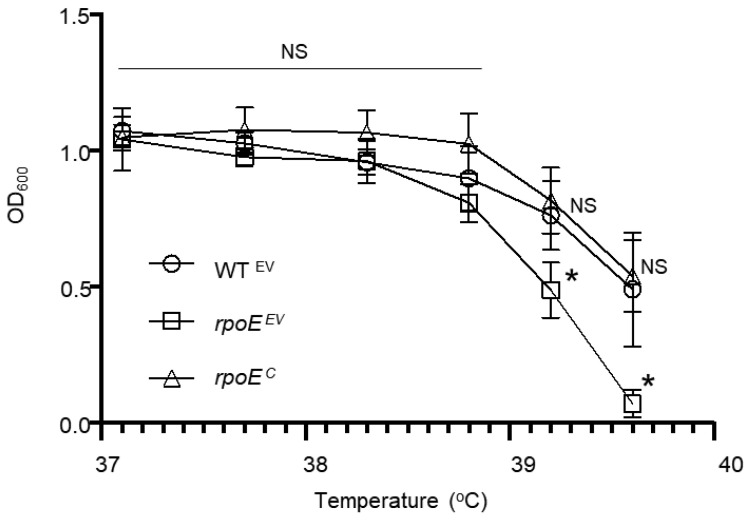
**Mutation of *rpoE* limits the growth potential for GAS at increasing temperatures.** Overnight cultures of WT 5448 (WT), *rpoE* mutant (*rpoE*), or the complemented strain (*rpoE^C^*) were diluted to a starting OD_600_ = 0.05 in fresh THY media; 120 µL of culture was then aliquoted into sterile PCR tubes and incubated in a thermocycler at increasing temperature gradient for 24 h and growth measured by absorbance (OD_600_). This experiment was performed in six biological replicates. Asterisks indicate statistical significance (*p* ≤ 0.05) at indicated growth temperatures of the *rpoE* mutant by comparison to the WT as determined by an unpaired *t*-test. NS = not significant.

**Figure 5 microorganisms-10-01686-f005:**
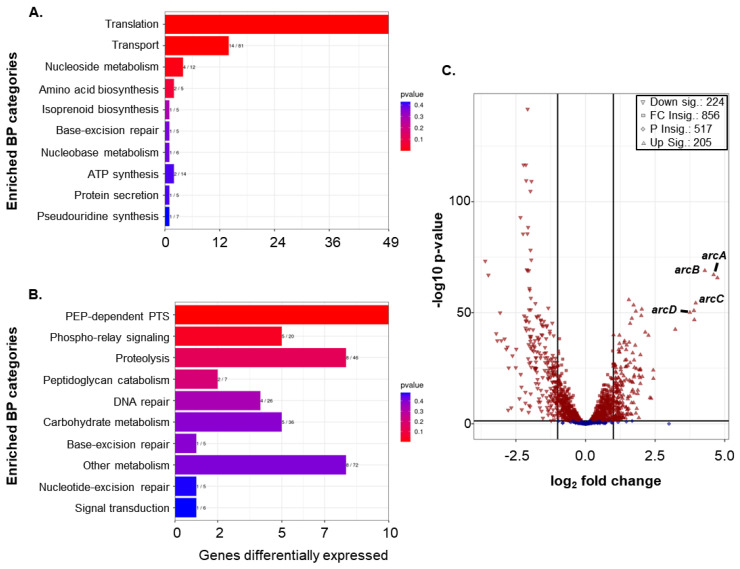
**RNA-seq analysis of the *rpoE* mutant compared to WT 5448 in THY medium.** Transcript levels were measured by RNA-seq for the *rpoE* mutant relative to the WT 5448 and ontology enrichments were analyzed. Genes that were upregulated (**A**) and those that were downregulated (**B**) were plotted based on biological purpose (BP) and in order of confidence based on their predicted BP using goseq software, with those groups most overrepresented being shown in red. (**C**) Volcano plot of RNA-seq DESeq2 analysis used to determine genes that were differentially expressed (log2) and *p*-value (log10). All genes greater than two-fold differential expression in the *rpoE* mutant vs. the WT are shown outside of the vertical black bars. Red symbols in the volcano plot represent genes that show a differential expression with a *p*-value < 0.05 whereas blue indicates those above that level.

**Figure 6 microorganisms-10-01686-f006:**
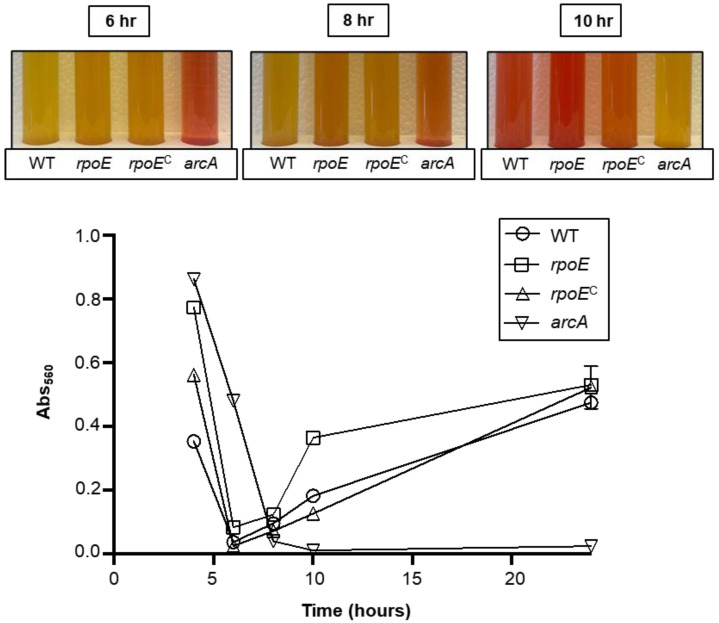
**Mutation of *rpoE* enhances the rate of the deacidification of the extracellular media.** Wild type (WT), the *rpoE* mutant (*rpoE*), its complemented variant (*rpoE*^C^), or an *arcA* mutant (*arcA*) cultures were standardized to a starting OD_600_ = 0.05 and were grown in THY supplemented with phenol red (18 µg mL^−1^) and supplemental arginine (30 mM). Time points were established based on preliminary studies. Color changes (**top**) and absorbance changes (**bottom**) in media pH over time are shown above. The absorbance of spent media was measured at 560 nm (Abs_560_) at the indicated time points. Each experiment is representative of three biological replicates.

**Table 1 microorganisms-10-01686-t001:** Bacterial plasmids and strains used in this study.

Plasmid or Strain	Description	Reference
**Plasmid**		
pMSP3535	Derived from pAMβ1 with a ColE1 replicon, Erm^R^, with nisin inducible system (*nisRK* P*nisA*)	[30]
pOri253	Derived from pIL253 (5.2 kb), 0.89-kb fragment of oriColE1 from pBluescript II KS	[29]
pP*rpoE*_*rpoE*	Derived from pOri253 and 1152 bp fragment of *rpoE* with ~400 bp of upstream intergenic DNA	This work
**Strain**		
5448 WT	Parental strain	[26]
5448 EV	Wild type with empty vector pMSP3535	This work
5448 *rpoE*	*rpoE*::pSinS mutant	This work
5448 *rpoE*^EV^	*rpoE*::pSinS mutant with empty pMSP3535 vector	This work
5448 *rpoE*^C^	*rpoE*::pSinS mutant with pP*rpoE*_*rpoE* complementation vector	This work
5448 *arcA*	*arcA*::pSinS mutant	[27]
5448 Spy_1849	Spy_1849::pSinS mutant	[28]
5448 *covS*	*covS*::pSinS mutant	[28]

## Data Availability

Raw sequencing reads were deposited in the Sequence Read Archive (SRA) at the National Center for Biotechnology Information (NCBI) under accession number PRJNA860170.

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
