# Peer review of "Loss of rpoE Encoding the δ-Factor of RNA Polymerase Impacts Pathophysiology of the Streptococcus pyogenes M1T1 Strain 5448"

_microorganisms, 2022, doi:10.3390/microorganisms10081686_

Round 1

Reviewer 1 Report

Microorganisms-1849110

Loss of rpoE encoding the δ-factor of RNA polymerase impacts pathophysiology of the Streptococcus pyogenes M1T1 strain5448

This manuscript is about investigation of novel role of rpoE encoding the δ-factor of RNA polymerase in pathogenicity of Streptococcus pyogenes. A mutant strain of rpoE was applied for the study of survival effect in a murine soft-tissue infection model and in vitro culture in human blood. The mutant strain was also applied for morphological test and temperature effect test.

Especially, RNA-seq analysis was performed to know the functional effect of rpoE, and several genes (such as arcA) are identified as differentially expressed genes. These results suggest rpoE genes are essential for pathogenicity of Streptococcus pyogenes. However, several comments (esp. RNA-seq. analysis) should be answered for insufficient description for this study.   

Major points

1)    You should indicate the replication number of RNA-seq experiments.

2)    Why did you use DESeq2, EdgeR, and Limma all (Line 200)? In general, only one of these is used for DEG analysis. Then, how did you compare them? I could not find the data related to this result.

3)    In this study, the vector-mediated mutation was conducted and rpoE gene was substituted by a partial gene of rpoE gene. Since the substitution of a gene will not cause a difference in expression level, the expression level of rpoE in both conditions should be similar in a well-controlled experiment. But in supplementary data, the substituted rpoE gene was decreased. How can you prove that the RNA-seq experiment was controlled well? or Can you explain why the expression level of rpoE gene was decreased?

4)    Generally, the enrichment test uses p-value 0.1 or less or q-value, but it seems an extremely high cut-off value (above 0.4) was applied in your analysis as shown in Figure 5. Nevertheless, the results of enrichment test did not show any relationship with the previous results (Figure1 to 3). Moreover, as described in Line 485-486, the overexpression in ADI pathway, which is the main result of RNA-seq, cannot fully explain pathophysiological changes. That is, RNA-seq results cannot explain the reason for altered virulence or fitness changes. The result of RNA-seq should be analyzed deeper to explain those changes.

5)    Figure 5 should be carefully corrected. For example, labels in the bar plots (Figure 5AB) are too small and the x-axis is out of the range. In the volcano plot (Figure 5C), identified DEGs are incorrectly labeled. Only genes that are outside of the vertical black bars should be colored with red. It also has an invisible legend box on the top right.

Minor points

1) The formats such as ‘from nlm.’, ‘Research Support’ in the reference should be corrected.

2) Line 31. ‘Group a Streptococcus’ should be ‘Group A Streptococcus’

3) Line 95. CovR in this paragraph seems to be CovS. It should be checked

4) Error bars in Figure 6 line graph will be needed.

5) The location of legend of Figure 6 should be corrected in the page 11

6) The meaning of paragraph ‘Qualitative changes (top) and quantitative changes (bottom) in

media pH over time are shown above’ in line 405 is obscure.

Author Response

Reviewer #1 Responses

This manuscript is about investigation of novel role of rpoE encoding the δ-factor of RNA polymerase in pathogenicity of Streptococcus pyogenes. A mutant strain of rpoE was applied for the study of survival effect in a murine soft-tissue infection model and in vitro culture in human blood. The mutant strain was also applied for morphological test and temperature effect test.

Especially, RNA-seq analysis was performed to know the functional effect of rpoE, and several genes (such as arcA) are identified as differentially expressed genes. These results suggest rpoE genes are essential for pathogenicity of Streptococcus pyogenes. However, several comments (esp. RNA-seq. analysis) should be answered for insufficient description for this study.   

Major points

  • You should indicate the replication number of RNA-seq experiments.

We indicate on line 350 that RNA was extracted from each strain of GAS in four separate biological replicates.

  • Why did you use DESeq2, EdgeR, and Limma all (Line 200)? In general, only one of these is used for DEG analysis. Then, how did you compare them? I could not find the data related to this result.

This was a typo and we have corrected the manuscript and removed irrelevant references. From line 199, the manuscript now reads: “Differential expression analyses were performed with DESeq2 [37], and a statistically uninformed basic method as negative control.”

  • In this study, the vector-mediated mutation was conducted and rpoE gene was substituted by a partial gene ofrpoE  Since the substitution of a gene will not cause a difference in expression level, the expression level of rpoE in both conditions should be similar in a well-controlled experiment. But in supplementary data, the substituted rpoE gene was decreased. How can you prove that the RNA-seq experiment was controlled well? or Can you explain why the expression level of rpoE gene was decreased?

The reviewer raises an interesting point, that given the nature of the rpoE mutation, there would likely be initiation of rpoE transcription and partial mRNA expression. However, there are several possible explanations for the decreased expression of rpoE observed during RNA-seq for the mutant. Since the pSinS interruption occurs at the extreme 5’ end of the rpoE ORF, there is likely very little transcript upstream to compensate for the loss downstream of the large insertion due to polarity. Our RNA-seq analysis of differential expression only takes into account cDNA reads located within annotated open reading frames (ORFs). Also, the distance between the 5’ UTR and the terminator sequence caused by kilobases of pSinS plasmid would likely lead to rpoE mRNA instability. Either of these possibilities would lead to reduced rpoE reads in the mutant compared to WT during RNA-seq analysis.

4)    Generally, the enrichment test uses p-value 0.1 or less or q-value, but it seems an extremely high cut-off value (above 0.4) was applied in your analysis as shown in Figure 5. Nevertheless, the results of enrichment test did not show any relationship with the previous results (Figure1 to 3). Moreover, as described in Line 485-486, the overexpression in ADI pathway, which is the main result of RNA-seq, cannot fully explain pathophysiological changes. That is, RNA-seq results cannot explain the reason for altered virulence or fitness changes. The result of RNA-seq should be analyzed deeper to explain those changes.

We acknowledge that the RNA-seq results do not offer an obvious explanation for the attenuation of the rpoE mutant. Thus, we hypothesize that the overall change in gene expression itself is what causes rpoE mutant attenuation as opposed to a change in any single/limited set of pathways. Our conclusion is corroborated by findings from Xue and colleagues [18] who inactivated rpoE in Streptococcus mutans and found global changes in protein involved in stress would likely cause an overall reduction in fitness (line 496-499). We do not rule out the possibility that there are in fact specific gene/genes that account for the attenuation, but searching for these genes might be futile and therefore we feel is best left for a subsequent study.

  • Figure 5 should be carefully corrected. For example, labels in the bar plots (Figure 5AB) are too small and the x-axis is out of the range. In the volcano plot (Figure 5C), identified DEGs are incorrectly labeled. Only genes that are outside of the vertical black bars should be colored with red. It also has an invisible legend box on the top right.

We have increased the font size of the labels in Fig 5AB. We are unsure what the reviewer means by “the x-axis is out of the range” and we have reviewed that the arc DEGs are indeed labeled correctly. We have enhanced the resolution of the legend in Fig 5C.  Red in the volcano plot represents genes that show a differential expression with a p-value below 0.05 whereas blue indicates those above that level.  This is used to indicate all of the data, regardless of the fold change DE, that is statistically significant. We apologize that this was not included in the figure legend and that has now been corrected.  It is also possible to indicate those genes above or below log2 of 1.0 with read; however, we chose this color scheme to indicate the overall quality of the data.

Minor points

  • The formats such as ‘from nlm.’, ‘Research Support’ in the reference should be corrected.

We have corrected the references as suggested.

  • Line 31. ‘Group a Streptococcus’ should be ‘Group A Streptococcus’

We have made the appropriate revision.

  • Line 95. CovR in this paragraph seems to be CovS. It should be checked

We have made the appropriate revision.

  • Error bars in Figure 6 line graph will be needed.

Error bars are present but not always visible at most time points given such a slight variation between technical replicates. As stated in the text, this experiment is one representative biological replicate with multiple technical replicates.

  • The location of legend of Figure 6 should be corrected in the page 11

We have modified the legend per the reviewer’s recommendations.

6) The meaning of paragraph ‘Qualitative changes (top) and quantitative changes (bottom) in

media pH over time are shown above’ in line 405 is obscure.

We have clarified the description of the figure legend such that it now reads: “Color changes (top) and absorbance changes (bottom) in media pH over time are shown above. Absorbance of spent media was measured at 560 nm (Abs560) at the indicated timepoints.

Reviewer 2 Report

Rom and Breton et al submitted a manuscript entitled “ Loss of rpoE encoding the δ-factor of RNA polymerase impacts  pathophysiology of the Streptococcus pyogenes M1T1 strain  5448 The authors work on the organism Streptococcus pyogenes, also known as the Group A Streptococcus (GAS), is a Gram-positive bacterial pathogen of major clinical significance. In this study, they investigated the importance of the gene encoding the delta (δ) subunit of the GAS RNA polymerase, rpoE, for its impact on virulence during skin and soft-tissue infection. Authors used knockout and complementation strategy for  rpoE gene and found that mutant  was defective for survival in whole human blood and was attenuated for both disseminated lethality and lesion size upon mono-culture infection in mouse soft tissue. Furthermore, the mutant had reduced competitive fitness when co-infected with a wild-type mouse model. They also found that size and the ability to grow at higher temperatures were both affected when grown with nutrient-rich THY media. They used whole transcriptome analysis and found that 429 total genes (205 up-regulated, 224 down-regulated) were affected and further they verified their RNA-seq result phenotypically.

Overall study was conducted very carefully, and the introduction, methods, results, and discussion were sufficiently described and well presented. I do feel this manuscript is suitable for the publication in Microorganism journal.

Author Response

Reviewer #2 Responses

Rom and Breton et al submitted a manuscript entitled “ Loss of rpoE encoding the δ-factor of RNA polymerase impacts  pathophysiology of the Streptococcus pyogenes M1T1 strain  5448 The authors work on the organism Streptococcus pyogenes, also known as the Group A Streptococcus (GAS), is a Gram-positive bacterial pathogen of major clinical significance. In this study, they investigated the importance of the gene encoding the delta (δ) subunit of the GAS RNA polymerase, rpoE, for its impact on virulence during skin and soft-tissue infection. Authors used knockout and complementation strategy for  rpoE gene and found that mutant  was defective for survival in whole human blood and was attenuated for both disseminated lethality and lesion size upon mono-culture infection in mouse soft tissue. Furthermore, the mutant had reduced competitive fitness when co-infected with a wild-type mouse model. They also found that size and the ability to grow at higher temperatures were both affected when grown with nutrient-rich THY media. They used whole transcriptome analysis and found that 429 total genes (205 up-regulated, 224 down-regulated) were affected and further they verified their RNA-seq result phenotypically.

Overall study was conducted very carefully, and the introduction, methods, results, and discussion were sufficiently described and well presented. I do feel this manuscript is suitable for the publication in Microorganism journal.

We appreciate Reviewer #2’s comments and confidence in that this manuscript is ready for publication.